# Chemotherapy effectiveness in trial-underrepresented groups with early breast cancer: A retrospective cohort study

Ewan Gray[1]*, Joachim Marti[2], Jeremy C. Wyatt[3], David H. Brewster[4], Peter S. Hall[4], SATURNE advisory group[¶]

1 University of Manchester, Manchester, United Kingdom, 2 Centre for Primary Care and Public Health (Unisanté), University of Lausanne, Lausanne, Switzerland, 3 University of Southampton, Southhampton, United Kingdom, 4 University of Edinburgh, Edinburgh, United Kingdom

¶ Membership of the SATURNE advisory group is provided in the Acknowledgments.
* ewan.gray@manchester.ac.uk

**Data Availability Statement:** Data cannot be shared publicly because these routinely collected healthcare data are considered sensitive and

## Abstract

### Background

Adjuvant chemotherapy in early stage breast cancer has been shown to reduce mortality in a large meta-analysis of over 100 randomised trials. However, these trials largely excluded patients aged 70 years and over or with higher levels of comorbidity. There is therefore uncertainty about whether the effectiveness of adjuvant chemotherapy generalises to these groups, hindering patient and clinician decision-making. This study utilises administrative healthcare data—real world data (RWD)—and econometric methods for causal analysis to estimate treatment effectiveness in these trial-underrepresented groups.

### Methods and findings

Women with early breast cancer aged 70 years and over and those under 70 years with a high level of comorbidity were identified and their records extracted from Scottish Cancer Registry (2001–2015) data linked to other routine health records. A high level of comorbidity was defined as scoring 1 or more on the Charlson comorbidity index, being in the top decile of inpatient stays, and/or having 5 or more visits to specific outpatient clinics, all within the 5 years preceding breast cancer diagnosis. Propensity score matching (PSM) and instrumental variable (IV) analysis, previously identified as feasible and valid in this setting, were used in conjunction with Cox regression to estimate hazard ratios for death from breast cancer and death from all causes. The analysis adjusts for age, clinical prognostic factors, and socioeconomic deprivation; the IV method may also adjust for unmeasured confounding factors. Cohorts of 9,653 and 7,965 were identified for women aged 70 years and over and those with high comorbidity, respectively. In the ≥70/high comorbidity cohorts, median follow-up was 5.17/6.53 years and there were 1,935/740 deaths from breast cancer. For women aged 70 years and over, the PSM-estimated HR was 0.73 (95% CI 0.64–0.95), while for women with high comorbidity it was 0.67 (95% CI 0.51–0.86). This translates to a mean predicted benefit in terms of overall survival at 10 years of approximately3%

access is legally restricted. Data are available to any research group via application to NHS Scotland Public Benefit and Privacy Panel and Information Services Division of NHS Scotland (website: https://www.isdscotland.org/Products-and-Services/EDRIS/). Study ID for reference: 1516-0251.

**Funding:** This study was funded by the Chief Scientist Office (CSO, https://www.cso.scot.nhs.uk/), Scotland (Grant reference number: HIPS/16/26, grant holder PH). The funders had no role in study design, data collection and analysis, decision to publish, or preparation of the manuscript.

**Competing interests:** The authors have declared that no competing interests exist.

**Abbreviations:** ER, estrogen receptor; IV, instrumental variable; PSM, propensity score matching; RWD, real world data; RWE, real world evidence.

(percentage points) and 4%, respectively. A limitation of this analysis is that use of observational data means uncertainty remains both from sampling uncertainty and from potential bias from residual confounding.

## Conclusions

The results of this study, as RWD, should be interpreted with caution and in the context of existing and emerging randomised data. The relative effectiveness of adjuvant chemotherapy in reducing mortality in patients with early stage breast cancer appears to be generalisable to the selected trial-underrepresented groups.

## Author summary

### Why was this study done?

- Women aged 70 years and over and with other health conditions were largely excluded from participating in the clinical trials that established the efficacy of adjuvant chemotherapy in early breast cancer.

- An attempted trial for women aged 70 years and over was abandoned due to failure to recruit participants, and observational data are therefore the best available option to investigate the generalisability of chemotherapy effectiveness to trial-underrepresented groups.

### What did the researchers do and find?

- A retrospective cohort study was conducted using a population-based cancer registry with linkage to other routinely collected health data in Scotland.

- Propensity score matching and instrumental variable methods were used to estimate the effect of chemotherapy on breast cancer mortality and all-cause mortality, adjusting for differences in prognosis between those who received chemotherapy and those who did not.

- The average predicted benefit of chemotherapy was an additional 3 out of every 100 women surviving for 10 years for those aged 70 years and over, and an additional 4 out of every 100 for those with other health conditions.

### What do these findings mean?

- These results support the generalisability of treatment effectiveness estimates for adjuvant chemotherapy for early breast cancer to women aged 70 years and over and those with other health conditions.

- These results should be interpreted with appropriate caution as they are estimated from observational data and may be biased by residual confounding.

## Introduction

The use of adjuvant chemotherapy after surgical treatment of early breast cancer is a major contributor to the reduction in mortality from breast cancer over the last 3 decades. A global collaboration of trialists published a definitive individual patient data meta-analysis of more than 100,000 women with breast cancer, concluding that chemotherapy reduces the risk of dying from breast cancer by about a third [1,2]. However, the clinical trials upon which this evidence relies were performed in highly selected patient populations including few patients older than 70 years or patients with other significant health conditions. In routine clinical practice there are many patients who would never have been included in those trials due to advanced age, comorbidity, or frailty. The decision of whether or not to administer or undergo adjuvant chemotherapy is informed solely by evidence from the 'trial eligible' patient population. Therefore, decisions are based on the assumption that the estimated treatment effect is generalisable, despite differences in personal characteristics.

A lack of evidence of generalisability of clinical trial results (the problem of external validity) has been recognised as a major barrier to translating research findings into changes in clinical practice [3]. Generalisability of experimental findings to the populations seen in clinical practice can be maximised by designing trials to be pragmatic, having wide inclusion criteria for patients, recruiting patients into trials from a variety of typical clinical settings, reducing differences between trial protocols and clinical practice, and obtaining data on relevant outcomes or adverse events [4].

Attempts to address a perceived lack of generalisability in the evidence base for adjuvant chemotherapy in patients aged 70 years and over by conducting further randomised controlled trials have failed due to poor recruitment [5,6]. In the pilot phase of the ACTION trial, a lack of equipoise on the part of both clinicians and patients was noted as the major reason for an unwillingness to participate in randomisation. The persistent lack of direct trial evidence for women aged 70 years and over engenders considerable uncertainty about the balance of patient benefit and harm from chemotherapy, which may lead to suboptimal treatment decisions and unwarranted variation in practice. When randomised studies are infeasible, as may be the case here, alternative methods using observational data may represent the best available source of evidence on treatment effectiveness. Observational data from routine sources have the potential to enhance external validity but at a cost of additional potential bias arising from the research design [7]. A lack of randomisation means that unaccounted for differences between patients who receive treatment and those who do not may bias results, a feature called residual confounding.

Prior analysis of Scottish Cancer Registry data demonstrated that several real world evidence (RWE) methods utilising available routine data from otherwise healthy women under 70 years are feasible and may give comparable results to randomised data in estimating the effectiveness of chemotherapy in early stage breast cancer [8]. Hazard ratios for breast cancer mortality in the trial-represented population were concordant between RWE and a randomised trial meta-analysis. However, results for all-cause mortality were less concordant, indicating a greater potential for bias in relation to this outcome [8].

This study aims to estimate the effectiveness of adjuvant chemotherapy for early stage breast cancer in reducing mortality for women aged 70 years and over and for women with a high level of comorbidity using real world data (RWD). The estimates are presented for consideration alongside available evidence from the trial-represented population.

## Methods

A retrospective cohort study design was used. All records of women with primary invasive breast cancer (ICD-10 C50) diagnosed from 1 January 2001 to 31 December 2015 were

retrieved from the Scottish Cancer Registry. Linkage to routine outpatient and inpatient records (ISD Scotland datasets SMR00 and SMR01, 1996 to 2017) was achieved using each patient's uniquely identifying Community Health Index (CHI) number. Selection and linkage was provided by ISD Scotland. Use of these anonymous data in this research project was reviewed and approved by the NHS Scotland Public Benefit and Privacy Panel. Follow-up of vital status was available to April 2017. Women with first breast cancer were identified based on the first chronological record of diagnosis code ICD-10 C50 for the unique patient identifier. In the case of multiple simultaneous records, the record with the most complete data was selected. If completeness was identical then the record with the worse prognosis (PREDICT score) was selected. When records were identical in all extracted variables, the duplicate records were deleted.

Exclusion criteria included male sex, advanced cancer (clinical M stage = 1), no recorded surgery, or recorded neoadjuvant therapy (chemotherapy or hormone therapy prior to surgery). PREDICT (version 2) prognostic scores [9] were estimated for all patients with complete input data. The prognostic algorithm has previously been shown to be well calibrated in this population [10]. Further details of the dataset and variables are available in [8].

The ≥70 patient group was selected based on age at date of diagnosis being 70 years or greater. Based on clinical expert opinion, high comorbidity was defined as meeting 1 or more of 3 conditions: (1) a score of 1 or more on the Charlson comorbidity index [11] based on inpatient records from the previous 5 years, (2) total inpatient bed days in the previous 5 years in the top decile (6 or more) of the full cohort, and/or (3) 5 or more outpatient visits to respiratory, cardiology, or rheumatology specialties in the previous 5 years. Women aged 70 years and over were excluded from the high comorbidity group. A sensitivity analysis selected women aged 70 years and over excluding those with high comorbidity.

## Statistical analysis

The choice of statistical analysis was determined based on prior assessment of the feasibility and validity of a range of econometric methods in the trial-represented population of cases from the same registry [8]. The study proposal for data analysis is included in S1 Text. The plan originally included only the regression discontinuity design but was later expanded to include regression adjustment, propensity score matching (PSM), and instrumental variable (IV) designs.

Two RWE designs—PSM and IV—were used to obtain estimates of adjuvant chemotherapy effectiveness. PSM was conducted using a propensity score as constructed in [8] with 1:1 nearest-neighbour matching within callipers, without replacement. Propensity scores were estimated using probit regression with the following explanatory variables: PREDICT 10-year probability of mortality, age at diagnosis, number of positive lymph nodes, pathological tumour size, tumour histological grade, mode of detection, estrogen receptor (ER) status, HER2 status, hormone therapy use, radiotherapy use, year of diagnosis, Scottish Index of Multiple Deprivation (SIMD) quintile, Charlson comorbidity index, log total inpatient bed days (in the 5 years prior to diagnosis), and log total outpatient visits (in the 5 years prior to diagnosis). Interactions of other clinical prognostic factors with ER status were also included. Two versions of the IV approach were estimated using (1) PREDICT chemotherapy benefit score and (2) PREDICT chemotherapy benefit score interacted with a post-2010 dummy variable. Two-stage residual inclusion was used to implement the IV approach as this is suitable when both the treatment and the outcomes are limited dependent variables [12]. Confidence intervals were calculated by simple bootstrap with 1,000 replications. Full details and justification for the selection of these RWE designs and specifications as the most suitable means of

providing estimates of adjuvant chemotherapy are available in [8] and S2 Text. The analyses were repeated for the outcomes of breast cancer mortality and all-cause mortality. Breast cancer mortality was defined as a breast cancer code recorded as the primary cause of death or 1 of the 3 contributing causes of death in the death certificate.

Directly comparable estimates of adjuvant chemotherapy effectiveness are taken from the Early Breast Cancer Trialists' Collaborative Group (EBCTCG) meta-analysis. The EBCTCG meta-analysis estimated that the HR for mortality from breast cancer for newer anthracycline regimens versus placebo was 0.71 (95% CI 0.62 to 0.83) [2]. The corresponding HR for all-cause mortality was 0.83 (95% CI 0.73 to 0.94).

HRs were estimated for comparison with trial meta-analysis reports, but HR is very limited for informing clinical decisions. Therefore, we have applied the estimated HR in a recalibrated version of the PREDICT model to produce an estimate of survival benefits (absolute risk reductions) over 10 years for all women in the sample for each group. This analysis was added following reviewer comments to aid clinical interpretation. Recalibration replaced the coefficient for chemotherapy use with a coefficient corresponding to the PSM-estimated HR for each group in this study. Expected benefit estimates were stratified in 5-year age bands for women aged 70 years and over, while women with comorbidities were stratified into 3 groups in 10-year age bands from 40 to 69 years. For the high comorbidity group, an additional hazard of non-breast-cancer death (HR ranging from 1–5) was applied to reflect how additional mortality from specific comorbidities might impact chemotherapy benefits (a method similar to that previously used in the [now defunct] Adjuvant Online! decision tool, in which the clinician could specify an additional risk of mortality [13]). Mean benefit from chemotherapy and the proportion of women with benefit at or above the guideline thresholds of 3% and 5% [14] were calculated.

## Results

A total of 9,653 and 7,965 eligible patient records were identified for the ≥70 group and the high comorbidity group, respectively (Fig 1). Patient characteristics in the ≥70 and high comorbidity groups are displayed in Table 1.

The results of PSM sample balance tests and IV first-stage results are available in S2 and S3 Tables. Good balance was demonstrated for both of the matched samples, with no clinically important differences in important covariates between treated and untreated groups. First-stage results for the IV analysis indicated statistically significant effects of the proposed instruments on the probability of receiving the treatment (≥70 IV1: 0.49 [95% CI 0.37–0.6], $P < 0.001$; IV2: 0.14 [95% CI 0.04–0.23], $P = 0.006$; high comorbidity IV1: 0.41 [95% CI 0.26–0.56], $P < 0.001$; IV2: 0.1 [95% CI 0.02–0.18], $P = 0.012$), an important assumption of the method. The estimated hazard ratios for death in women aged 70 years and over and women with high comorbidity are displayed in Table 2.

Breast cancer mortality hazard ratios for women aged 70 years and over and those with high comorbidity are consistent with trial meta-analysis estimates. There was a closer match with the PSM estimates and somewhat lower HRs reported using the IV method. The confidence intervals indicate that sufficient uncertainty remains such that identical HRs between any of these methods and the trial meta-analysis cannot be ruled out at conventional thresholds, i.e., there was no strong evidence against generalisability. Overall results indicate a beneficial effect for chemotherapy at conventional statistical thresholds with the exception of IV1 for breast cancer death and IV1 and IV2 for all deaths in the high comorbidity group. Results were not sensitive to excluding women with high comorbidity from the ≥70 group (S4–S6 Tables).

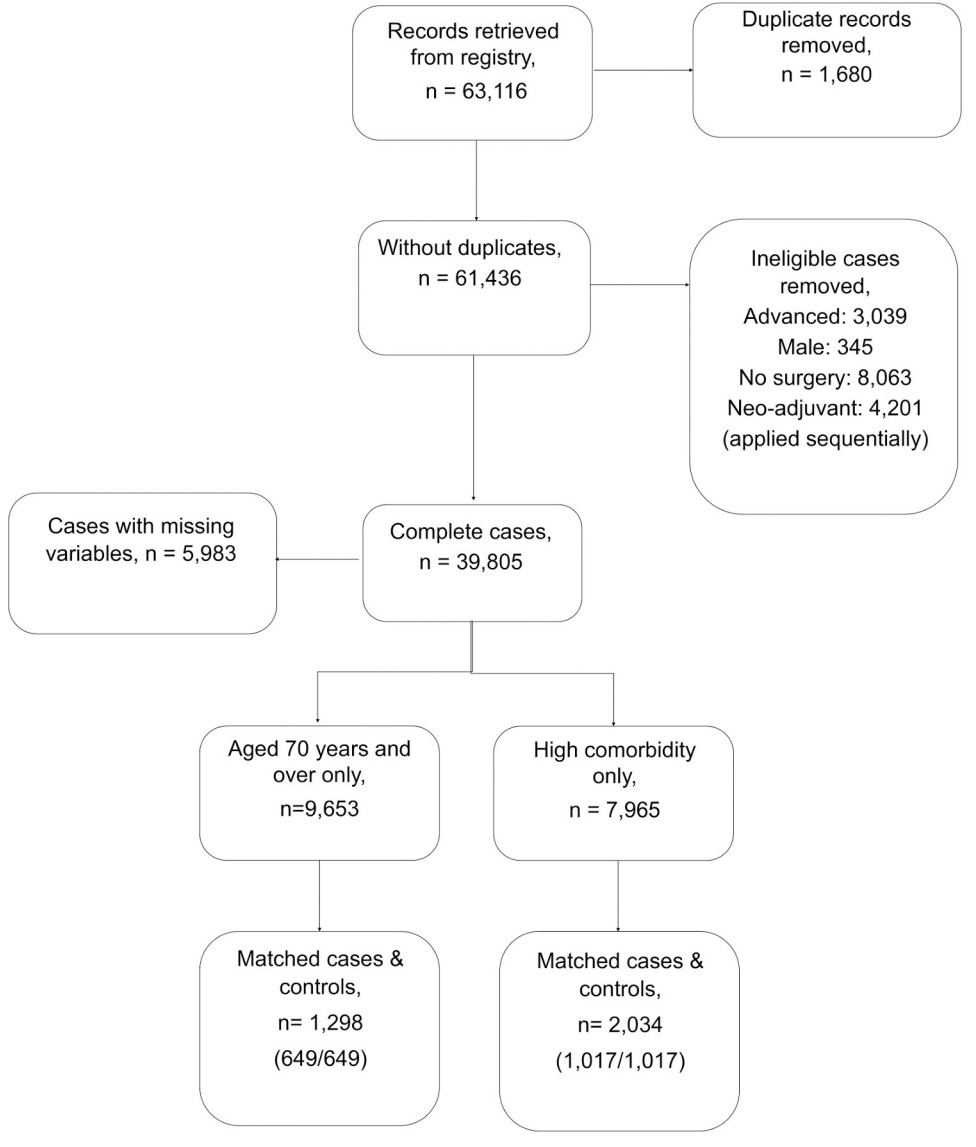

**Fig 1. Patient record selection.**

Tables 3 and 4 translate these results into clinically meaningful estimates of overall survival benefit using a recalibrated version of the PREDICT model. For women aged aged 70 years and over, predicted benefits over 10 years average around a 3-percentage-point increase in survival. Benefits of 3% and above were estimated for approximately 40% of women of any age, while benefits of 5% and above were estimated for approximately 20% of women up to the age of 85 years. Over the age of 85 years, there are few women with predicted benefits at or above the 3% threshold and almost none at or above the 5% threshold. The actual proportion of women aged 70 years and over who received chemotherapy was 10.2%.

In younger age groups with high comorbidity, the benefits are commensurably greater due to less age-related competing risk of mortality from non-breast-cancer causes. Predicted survival benefit from chemotherapy is related to the assumed increased hazard of mortality from other causes, but only weakly. Even with a relatively large additional risk of mortality from comorbidity (HR = 5), the mean absolute benefit and proportion of women at or above the 3%

**Table 1. Summary statistics of trial-underrepresented group samples from the Scottish Cancer Registry, 2001–2015.**

| Characteristic | ≥70 years | High comorbidity |
|---|---|---|
| Total number of patients | 9,653 | 7,965 |
| Total time at risk (years) | 56,864 | 57,094 |
| Median follow-up (years) | 5.17 | 6.53 |
| Number of breast cancer deaths | 1,935 | 740 |
| Number of other deaths | 2,018 | 648 |
| Five-year survival rate | 74.7% | 87.2% |
| Median age at diagnosis, years | 76 | 60 |
| Age | 76.65 (0.05) | 58.75 (0.14) |
| Tumour size (mm) | 24.68 (0.16) | 19.79 (0.21) |
| Inpatient days (in the 5 years prior) | 4.99 (0.15) | 13.39 (0.45) |
| PREDICT benefit score | 3.02 (0.02) | 2.68 (0.04) |
| Outpatient visits (in the 5 years prior) | 7.22 (0.10) | 12.68 (0.25) |

Data given as mean (SD) unless otherwise indicated. Additional summary statistics available in S1 Table.

and 5% thresholds are largely preserved. The actual proportion of women under 70 years with high comorbidity who received chemotherapy was 32.1%.

## Discussion

These observational data suggest that the relative effectiveness of adjuvant chemotherapy in reducing mortality in women with early stage breast cancer appears similar for trial-underrepresented groups (aged 70 years and over and high comorbidity) and trial-eligible groups. If one accepts that the additional assumptions required by RWE methods are met in this case, then this would imply that estimates of treatment effectiveness among trial-eligible patients are generalisable to these trial-underrepresented groups. These results are also in agreement with previous studies using observational data from the Surveillance, Epidemiology, and End Results (SEER) database for women aged 65 years and above [15].

### Limitations

The main limitation relating to this analysis is that, despite use of the more robust RWE methods for causal inference, there remains potential for bias in these results from residual confounding [7]. Residual confounding may arise from lack of data regarding chemotherapy regimens, limited use of the full dimensionality of the comorbidity data, or from as yet unknown confounding variables. For this reason, these results are not a direct substitute for evidence from a high-quality randomised trial. Data included in this study are from a single country, which may limit generalisability to other settings.

### Strengths

A strength of this study is the use of a large population-based cohort of patients with high-quality outcome determination and covariate information obtained from data linkage. Previous studies using SEER data did not have access to the same range of covariates. To our knowledge, no previous study has tested the feasibility of alternative methods of analysis to the same extent as was undertaken in this research. Use of a population-based cohort enhances external validity, allowing better assessment of effectiveness in clinical practice as opposed to efficacy in a highly selected trial population.

**Table 2. Hazard ratios for death from breast cancer and all causes in women aged 70 years and over and women with high comorbidity.**

| Real world evidence method | N | Breast cancer death | | | Death from any cause | | |
|---|---|---|---|---|---|---|---|
| | | Deaths | HR | 95% CI | Deaths | HR | 95% CI |
| Reference[1] | | | 0.71 | 0.62–0.83 | | 0.83 | 0.73–0.94 |
| Women aged ≥70 years | | | | | | | |
| PSM | 1,298 | 431 | 0.78 | 0.64–0.95 | 568 | 0.71 | 0.60–0.85 |
| IV1 | 9,653 | 1,935 | 0.57 | 0.42–0.74 | 3,953 | 0.61 | 0.49–0.74 |
| IV2 | 9,653 | 1,935 | 0.57 | 0.42–0.73 | 3,953 | 0.63 | 0.50–0.76 |
| Women with high comorbidity | | | | | | | |
| PSM | 2,034 | 254 | 0.67 | 0.51–0.86 | 421 | 0.67 | 0.56–0.82 |
| IV1 | 7,965 | 740 | 0.68 | 0.42–1.10 | 1,388 | 0.92 | 0.63–1.33 |
| IV2 | 7,965 | 740 | 0.59 | 0.37–0.99 | 1,388 | 0.82 | 0.58–1.22 |

[1]Early Breast Cancer Trialists' Collaborative Group meta-analysis of newer anthracycline-containing regimens versus placebo [2].

IV, instrumental variable; PSM, propensity score matching.

## How this study can inform future research

An important issue that this study informs is whether or not additional randomised trials are needed in the trial-underrepresented groups we have identified. Our results suggest that trials conducted within these groups with a no-chemotherapy arm are probably neither necessary nor desirable. A beneficial treatment effect, consistent with the reported trial meta-analysis, was observed in both groups. While previous attempts to conduct trials in women aged 70 years and over proved infeasible due to lack of recruitment, our results may change the clinical community's interpretation of the existing evidence and willingness to recruit such women to trials comparing 2 forms of therapy, although patients' perception of the risks and benefits and willingness to participate may remain a barrier. Also, some additional randomised data may become available in more selected populations. Randomisation is being conducted among patients classified into specific risk groups in ongoing studies [16,17], and standard adjuvant chemotherapy is a control arm in trials of new targeted therapies [18,19]. These trials will provide some new randomised data in the ≥70 age group, as the age restrictions are more relaxed in most of these trials than was the case in previous studies. However, the total number of patients randomised in this group may still be small.

**Table 3. Predicted survival benefit with chemotherapy for women aged 70 years and over.**

| Age | N | Predicted 10-year survival without chemo (%) | Predicted 10-year survival with chemo (%) | Mean absolute mortality benefit (%)* | Proportion with ≥3% benefit (%) | Proportion with ≥5% benefit (%) | Observed actual percent that received adjuvant chemo |
|---|---|---|---|---|---|---|---|
| 70–74 | 3,955 | 57.9 | 62.x | 2.9 | 39.6 | 19.2 | 18.2 |
| 75–79 | 3,131 | 45.8 | 49.6 | 3.1 | 44.2 | 18.9 | 7.3 |
| 80–84 | 1,784 | 32.x | 35.4 | 3.x | 46.4 | 17.3 | 1.5 |
| 85–99 | 681 | 17.3 | 20.4 | 3.x | 48.8 | 9.7 | 0.6 |
| 90–95 | 95 | 6.x | 8.6 | 2.7 | 43.2 | 0 | 0 |

*Percentage point improvement in survival at 10 years.

chemo, chemotherapy.

**Table 4. Predicted survival benefit with chemotherapy for women aged under 70 years with high comorbidity.**

| Age and additional hazard of non-BC death | N | Predicted 10-year survival without chemo (%) | Predicted 10-year survival with chemo (%) | Mean absolute mortality benefit | Proportion with ≥3% benefit (%) | Proportion with ≥5% benefit (%) | Observed actual percent that received adjuvant chemo |
|---|---|---|---|---|---|---|---|
| **Age 40–49 years** | 673 | | | | | | 62.2 |
| 1 | | 73.4 | 81.3 | 5.1 | 61.7 | 41.5 | |
| 1.5 | | 72.3 | 80.1 | 5.1 | 61.4 | 41.2 | |
| 2.5 | | 70.1 | 77.7 | 5.x | 60.3 | 40.3 | |
| 5 | | 64.9 | 72.2 | 4.7 | 59.7 | 38.6 | |
| **Age 50–59 years** | 2,991 | | | | | | 36.3 |
| 1 | | 79.4 | 84.1 | 3.4 | 37.4 | 24.5 | |
| 1.5 | | 77.1 | 81.7 | 3.3 | 37.1 | 23.9 | |
| 2.5 | | 72.7 | 77.1 | 3.2 | 36.6 | 23.2 | |
| 5 | | 62.8 | 67.x | 3.1 | 35.8 | 21.3 | |
| **Age 60–69 years** | 3,793 | | | | | | 23.1 |
| 1 | | 73.2 | 77.2 | 3.3 | 36.4 | 22.5 | |
| 1.5 | | 68.4 | 72.3 | 3.2 | 35.7 | 21.9 | |
| 2.5 | | 59.8 | 63.5 | 3.x | 35.1 | 20.6 | |
| 5 | | 42.9 | 46.3 | 2.9 | 34.2 | 18.8 | |

BC, breast cancer; chemo, chemotherapy.

A recommendation for future research is that the RWE estimates produced here should be replicated using comparable routinely collected data from other regions or countries. This is an important step in validating the results and could help to identify any biases arising from measurement or selection in these specific routine datasets. As RWD are not gathered solely for the purposes of research, the measurement and recording of some variables may be suboptimal in any given RWD [20]. One advantage of RWD is that replication is relatively inexpensive and does not raise additional ethical concerns related to equipoise, in contrast to replication of a randomised trial. Following replication of our results, a careful synthesis of both observational and trial data results should be attempted. This must reflect all the available data as well as current beliefs about treatment effect generalisability in this context. An important future direction for RWE will be to undertake more specific exploratory analysis to look for predictors of outcome within trial-underrepresented groups. This type of analysis will be useful to inform targeted data collection within specific underserved or at-risk groups.

### How this study can inform clinical practice

This analysis can inform treatment guidelines and patient information in trial-underrepresented groups—as exemplified in our presentation of expected benefits in terms of absolute survival using a recalibrated decision tool. Additional methodological development is also needed to support this type of evidence synthesis. First, more clarity is needed around the best methods for synthesis of observational and randomised data [21]. Second, methods must also address the generalisability of effectiveness estimates beyond the trial population, based on both extrapolation from existing data and prior beliefs based on other knowledge of the topic of study.

Evidence from observational data using RWE methods supports the generalisability of treatment effectiveness estimates for adjuvant chemotherapy for early breast cancer to women aged 70 years and over and those with high levels of comorbidity. The results of this study, as with all RWE, should be interpreted with appropriate caution and in the context of existing and emerging randomised data.

## Supporting information

**S1 RECORD Checklist.**
(DOCX)

**S1 Table. Additional summary statistics of trial-underrepresented group samples.**
(DOCX)

**S2 Table. PSM sample covariate balance for women aged 70 years and over and women with high comorbidity.**
(DOCX)

**S3 Table. First-stage regression results for each specification for women aged 70 years and over and women with high comorbidity.**
(DOCX)

**S4 Table. PSM estimates of adjuvant chemotherapy effectiveness for women aged 70 years and over excluding those with high comorbidity.**
(DOCX)

**S5 Table. First-stage regression results for each specification for women aged 70 years and over excluding those with high comorbidity.**
(DOCX)

**S6 Table. IV results for women aged 70 years and over excluding those with high comorbidity.**
(DOCX)

**S1 Text. Chemotherapy effectiveness in trial-underrepresented groups with early breast cancer: A retrospective cohort study—relevant sections of study proposal for data analysis.**
(DOCX)

**S2 Text. Details of PSM and IV methods.**
(DOCX)

## Acknowledgments

We would like to thank the SATURNE advisory group—David Cameron, Fiona Watt, Iain MacPherson, Larry Hayward, Colin McCowen, Gianluca Baio, and Paul Pharoah—for their generous advice and support.

## Author Contributions

**Conceptualization:** Joachim Marti, Jeremy C. Wyatt, David H. Brewster, Peter S. Hall.

**Data curation:** Ewan Gray.

**Formal analysis:** Ewan Gray.

**Funding acquisition:** Joachim Marti, Jeremy C. Wyatt, David H. Brewster, Peter S. Hall.

**Methodology:** Ewan Gray, Joachim Marti, Jeremy C. Wyatt, David H. Brewster, Peter S. Hall.

**Writing – original draft:** Ewan Gray.

**Writing – review & editing:** Ewan Gray, Joachim Marti, Jeremy C. Wyatt, David H. Brewster, Peter S. Hall.

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
