## [Decision Letter · Decision Letter 0]

30 Aug 2019

Dear Dr. Gray,

Thank you very much for submitting your manuscript "Real-world evidence for chemotherapy effectiveness in trial under-represented groups with early breast cancer" (PMEDICINE-D-19-02078) for consideration at PLOS Medicine. 

[LINK]

In light of these reviews, I am afraid that we will not be able to accept the manuscript for publication in the journal in its current form, but we would like to consider a revised version that addresses the reviewers' and editors' comments. Obviously we cannot make any decision about publication until we have seen the revised manuscript and your response, and we plan to seek re-review by one or more of the reviewers. 

We expect to receive your revised manuscript by Sep 20 2019 11:59PM. Please email us (plosmedicine@plos.org) if you have any questions or concerns.

We look forward to receiving your revised manuscript. 

Sincerely,

Adya Misra, 

Senior Editor 

PLOS Medicine

plosmedicine.org

Please to provide a url and accession number needed to apply for data access.

Please include a reporting checklist such as RECORD so we can suggest that.

http://www.equator-network.org/reporting-guidelines/record/ and provide the completed checklist as supplementary information. Please do not use page numbers as these are likely to change. 

The foornotes should be moved into the main text.

Please clarify where you received ethical approval, the committee should be listed in the Methods section.

Please revise your title according to PLOS Medicine's style. Your title must be nondeclarative and not a question. It should begin with main concept if possible. "Effect of" should be used only if causality can be inferred, i.e., for an RCT. Please place the study design ("A randomized controlled trial," "A retrospective study," "A modelling study," etc.) in the subtitle (ie, after a colon).

References-Please use the "Vancouver" style for reference formatting, and see our website for other reference guidelines https://journals.plos.org/plosmedicine/s/submission-guidelines#loc-references

Abstract-please clarify what is included in “high level of comorbidities”

Abstract-please combine methods and findings into a single sub-heading

Abstract-please include a sentence describing the limitations of your study at the end of the methods and findings section

Abstract-please provide further demographics of women included in this study within the methods/results

Abstract-Please quantify the main results (with 95% CIs and p values). 

Abstract-Please include the important dependent variables that are adjusted for in the analyses. 

Abstract-Please include the actual amounts and/or absolute risk(s) of relevant outcomes (including NNT or NNH where appropriate), not just relative risks or correlation coefficients. (example for absolute risks: PMID: 28399126)

"Did your study have a prospective protocol or analysis plan? Please state this (either way) early in the Methods section.

c) In either case, changes in the analysis—including those made in response to peer review comments—should be identified as such in the Methods section of the paper, with rationale."

Introduction-please simplify the last sentence in the first paragraph “When a patient is not themselves part of that population then decisions are based

on the assumption that the estimated treatment effect is generalisable, despite the differences

in personal characteristics”

Introduction-please provide a reference for “When randomised studies are infeasible, as in this case, alternative methods using observational data may represent the best available source of evidence for treatment effectiveness” or consider toning down

Introduction-please provide a reference for “Hazard ratios for breast cancer mortality in

the trial-represented population were concordant between RWE and a randomised trial metaanalysis. However, results for all-cause mortality were less concordant, indicating a greater potential for bias in relation to this outcome”

Introduction- please provide a reference for “Clinical expert opinion suggests that the predominant chemotherapy regimens in use in Scotland in this time period were anthracycline containing (CEF/CAF) with cumulative doses similar to those described in the more recent randomised studies” and for “The EBCTCG metaanalysis

estimated that the HR for mortality from breast cancer for newer anthracycline

regimens versus placebo was 0.71 (0.62 to 0.83) (2). The corresponding HR for all-cause

mortality was 0.83 (0.73 to 0.94)”

Table 1- please provide units of measurement for Tumour size

Results-Please clarify the sentence “Good balance was demonstrated for both of the matched samples”

Results-Please provide a p-value and 95% CI for “First-stage results for IV indicated statistically significant effects of the proposed instruments on the probability of receiving the treatment, an important assumption of the method” and “For breast cancer mortality hazard ratios for women aged over 70 were very similar to trial meta-analysis estimates for PSM and somewhat lower using IV”

Results- Please provide further details regarding “Results in relation to all-cause

mortality report lower hazard ratios compared to the corresponding trial meta-analysis

estimates, with the exception of IV1 and IV2 estimates in the high comorbidity group”. 

Results-please explain “degree of bias in all-mortality results may vary between over 70 and

high comorbidity groups reflecting differences in patient selection into treatment group” 

Comments from the reviewers:

Reviewer #1: This article reports that hazard ratios for breast-cancer specific survival observed using real world data (Scottish registry) in elderly/frail populations are somehow similar to that noted in randomized trials.

General comment:

Two aspects can be distinguished in that paper:

- The real-world data analysis "exercise", using different tools (propensity score matching and instrumental variable analyses), applied to a large dataset that has been linked to (some of the) patient characteristics. A very brief reminder on the principles/differences between these two analyses would be welcome for non-experts in the introduction or discussion, as the PLoS Medicine audience is much broader than real world data experts. Note that the discussion is focusing exclusively on the potential role of real-world evidence (with much emphasis), with no discussion on the medical relevance of the findings.

- The overall interest of these results for breast oncologists, which is very limited. There is no clinical data available in the main manuscript, no subgroup analysis... The bottom line of this study is that, in (the few?) patients that were considered as fit enough to receive chemotherapy, the observed hazard ratio for breast cancer-specific survival is apparently similar to that in randomized trials. However, this patient population has a much shorter lifespan, so the impact of chemotherapy on overall survival remains limited and cannot be the same than the one observed in younger/healthier patients (that are included in trials). So, it is satisfactory to know that Scottish oncologists were right when prescribing adjuvant chemotherapy in some patients from a highly selected elderly/frail population, but I don't see many other uses of these results. More clinically relevant results would be, e.g. the number needed to treat, classified by age (or by number/type of comorbidities) and focusing on OS rather than BCSS.

Reviewer #2: This paper investigates whether results on chemotherapy effectiveness for treating early breast cancer, derived from randomised trials, are generalisable to patients that are under-represented in these trials.

It tests this by deriving analogous hazard ratios from real world data using statistical methods that the authors have previously shown to be feasible in this context.

This contributes to an important area of research, aiming to determine the most effective individually appropriate and consistently administered treatment decisions for older women with early breast cancer who may also have comorbidities.

On the whole it is straightforward and clearly presented, particularly as the technicalities of the methods used are referenced elsewhere (in an earlier paper by the same authors) rather than described in detail.

This paper is something of a corollary to that earlier paper but nevertheless makes an valid contribution in respect to the trial unrepresented groups.

There are a few places where the meaning is not clear and I would suggest that the discussion in particular should be made to read in a more clear and focussed way.

Methods:

Complete case analysis was used, but there is no discussion of the possible implications of this. Was any consideration given to using multiple imputation, for instance, to complete the missing data?

Results and discussion:

I think the description and discussion of the results could be sharpened up considerably.

Phrases such as "very similar ... somewhat lower ... even more closely aligned" are a bit vague and possibly subjective if there is no predefined criteria of accuracy.

Simply stating the actual differences might be better. Discussing the confidence intervals, as is done, is more objective but needs to be done better for the high-comorbidity group.

There is also the point that by using 3 methods you increase the chance of at least one of them agreeing with the trial results. I say this in particular with reference to the authors' comment that the IV methods are more successful in the high-comorbidity group.

I would suggest it is more important that the three methods all clearly show a beneficial treatment effect in the case of the over 70 group, as shown by the confidence intervals.

For the high comorbidity group this becomes slightly more marginal, with one CI bordering on including unity and another clearly doing so. This should be noted and discussed.

I think this will still allow for the methods used to be proposed as potentially useful for assessing generalisability and will point more clearly to the need for future work to consolidate these methods.

Detailed comments:

p3 para2

Should this be "A lack of evidence of generalisability ..."

p4 last sentence of introduction

I'm not sure what this actually means in practice. Maybe just make the sentence simpler.

Table 2 and Table 3

Adding a row to both of these tables with the EBCTCG meta analysis HR values would make comparison easy and help the reader.

p8 sentence 3: "Point estimates ..."

Comparison is being made between the analysis for the two groups and also between the estimates for the high comorbidity group and trial values. It is not entirely clear therefore which of these the "greater uncertainty" refers to.

p8 final sentence of Results: "It should be noted ..."

I can make nothing of this sentence and it should be rewritten more clearly.

p8 first sentence of Results

".. appears similar .." is rather vague and non-committal

next sentence

Does it imply they are generalisable or rather "not provide evidence against generalisability"?

p9 "Our results suggest ... "

This sentence presumably relies on the fact that, regardless of the exact HR values, the results (nearly) all suggest a beneficial treatment effect. If so, make this explicit.

p9 "Although trials in ... "

Unclear what this sentence is saying

p9 I dislike the use of "has been/will be" 

p10 "This analysis should ultimately ..."

Use of "should" makes it sound unconvincing

p10 second para

This whole paragraph is poorly expressed. Especially the second sentence.

p11 Final paragraph

My judgment would be that this mostly supports the generalisability (of the fact that chemotherapy is effective) to these groups but the exact level of benefit is no so clearly generalisable.

Reviewer #3: This paper uses propensity matching and instrumental variables to test the effectiveness of adjuvant chemotherapy using routinely collected data. Overall, I found the paper interesting and easy to follow; however, I have some concerns especially around the lack of details in the methods and results.

Major comments:

* I realise that similar methods were used previously (reference # 6 by the same authors); however, the current manuscript would benefit from the inclusion of additional details in the statistical analysis section including the propensity score calculation, the analysis models, the prognostic ability of the used instrumental variable, etc. I have included more specific comments related to this issue below.

* I found the results section a bit too succinct. I believe additional results (additional outcomes, graphics, inclusion of tables currently in the supplement) would make the manuscript more interesting (see suggestions below).

* Please include relevant checklists in the supplement. See https://journals.plos.org/plosmedicine/article?id=10.1371/journal.pmed.1001885

Minor comments:

* Consider including the main estimates together with their confidence intervals in the abstract

* Consider using "routinely-collected data" instead of "real-world data"

* In the method section, please clarify what period served as the "baseline" to derive covariates and variables included in the propensity score calculation. Did the baseline period consist of the five years preceding the cancer diagnosis? For women with a diagnosis in 2001, I then assume that the linked data went as far back as 1996?

* If I understand correctly, the group of women over 70 could include women with high comorbidities but the high-comorbidity group excluded women over 70. Please confirm/clarify.

* Please clarify how potential controls for PSM were identified (i.e. eligibility). Please also clarify what the PREDICT chemotherapy benefit score consist of. For completeness, please also indicate the dependent variable used in the propensity score probit model (presumably treatment with adjuvant chemotherapy).

* Please clarify how the covariates included in the propensity score calculation were selected and what other variables were available (but not selected) in the linked data. Have the authors checked the balance after matching between other variables (those not included in the propensity score calculation)? Were the variables included in the propensity score calculation also used as covariates for the model analysing the outcome (principle of double-robustness?

* In footnote 2 on Page 5, "chemotherapy use" appears in the list of explanatory variables included in the propensity model. I would have thought that chemotherapy use would be the dependent variable, not an explanatory variable. Please correct/clarify.

* The last paragraph of the "statistical analysis" section which starts with "clinical expert opinion suggests…" appears to relate to background information rather than methods. I would suggest moving it to the introduction. 

* In the methods, please explain how the data was analysed including the models used, the methods for censoring data, any covariate adjustment, handling of missing data, potential adjustment for the within matched pair correlations, software used, etc..

* Please clarify/report on the prognostic ability of the PREDICT score. 

* Please consider including the tables showing the balance between the matched group in the main manuscript instead of in the supplement. It might also be interesting to see the balance (or lack thereof) before matching. In the "balance table" please clarify what "P NH: m1=m2" and "SMD" mean and how they were calculated e.g. 2-sample t-test.

* Figure 1. Please add information to describe the selection of matched controls.

* Please consider combining tables 2 and 3. Please make sure the number of decimals is consistent.

* Please consider adding Kaplan-Meier plots

* The last paragraph of the results placed between Table 3 and the discussion appears to be more about discussing and interpreting the results that about the reporting the results. Please consider integrating this paragraph into the discussion itself.

* I would like to see more discussion around the potential for residual confounding including reference to the variables available in the linked data but not included in the matching as well as variables not available at all.

Laurent Billot

[LINK]

---

## [Decision Letter · Decision Letter 1]

13 Nov 2019

Dear Dr. Gray,

Thank you very much for re-submitting your manuscript "Real-world evidence for chemotherapy effectiveness in trial under-represented groups with early breast cancer; a retrospective cohort study" (PMEDICINE-D-19-02078R1) for review by PLOS Medicine.

I have discussed the paper with my colleagues and the academic editor and it was also seen again by xxx reviewers. I am pleased to say that provided the remaining editorial and production issues are dealt with we are planning to accept the paper for publication in the journal.

[LINK]

We look forward to receiving the revised manuscript by Nov 20 2019 11:59PM. 

Sincerely,

Adya Misra, PhD

Senior Editor 

PLOS Medicine

plosmedicine.org

Requests from Editors:

D-19-02078

Title- please remove "real world evidence" 

DAS- please revise to “anonymously” Please to provide a url and accession number needed to apply for data access

Did your study have a prospective protocol or analysis plan? Please state this (either way) early in the Methods section.

c) In either case, changes in the analysis—including those made in response to peer review comments—should be identified as such in the Methods section of the paper, with rationale."

Abstract- Background refers to early stage breast cancer and the methods and findings section refers to primary. If both are the same, could we use consistent language. 

Square brackets for all references 

Footnotes- should be incorporated into main body of text

Discussion lines 22-24 this needs to be revised and toned down since the introduction states problems with recruitment of women over 70 were due to physicians and patients 

Discussion requires a subsection outlining the limitations of the study

Discussion section requires toning down of the conclusions as this is a retrospective cohort study not a trial. Please provide a brief explanation of how this study design allows you to infer the "effectiveness" versus "efficacy" of chemotherapy. 

RECORD checklist should be provided as a stand-alone SI file and page numbers should be removed as they are likely to change during publication

Comments from Reviewers:

Reviewer #3: I am comfortable with the responses and revisions. However, I am slightly confused by the new analyses reporting predicted survival benefits (Tables 3 and 4). In Table 3, the header of columns 3 and 4 state "Mean predicted 10-year survival ... (%)". Are these numbers showing the proportion of women expected to survive at 10 years? If so, I believe the word "mean" should not be there. If it is indeed a mean/average, then I do not understand what these numbers represent. For transparency, I would suggest indicating that these analyses were done post-hoc. The authors might add that they were conducted in response to the initial peer-review.

-Laurent Billot

[LINK]

---

## [Editor Report · Decision Letter 2]

26 Nov 2019

Dear Dr Gray, 

On behalf of my colleagues and the academic editor, Dr. Steven Shapiro, I am delighted to inform you that your manuscript entitled "Chemotherapy effectiveness in trial under-represented groups with early breast cancer: a retrospective cohort study" (PMEDICINE-D-19-02078R2) has been accepted for publication in PLOS Medicine. The publication date for your manuscript will be December 31, 2019. 

PRODUCTION PROCESS

PRESS

PROFILE INFORMATION

Thank you again for submitting the manuscript to PLOS Medicine. We look forward to publishing it. 

Best wishes, 

Adya Misra, PhD

Senior Editor 

PLOS Medicine

plosmedicine.org